# Factors promoting sustainability of NURHI programme activities in Ilorin and Kaduna, Nigeria: findings from a qualitative study among health facility staff

Adesola Oluwafunmilola Olumide [iD] ,[1,2] Courtney McGuire,[3] Lisa Calhoun,[3] Ilene Speizer,[4] Tolulope Babawarun,[2] Oladosu Ojengbede[2,5]

► Prepublication history and additional materials for this paper is available online. To view these files, please visit the journal online (http://dx.doi.org/10.1136/bmjopen-2019-034482).

For numbered affiliations see end of article.

**Correspondence to**
Dr Adesola Oluwafunmilola Olumide; daisyolu@yahoo.co.uk

## ABSTRACT

**Objectives** The Nigerian Urban Reproductive Health Initiative (NURHI) was implemented in six urban sites in Nigeria from 2009 to early 2015. Under a second phase (NURHI-2), activities ceased operations in four of the original six sites in 2015 (Ilorin, Abuja, Benin City and Zaria), and continued in two sites (Kaduna and Ibadan). This paper examines the sustainability of facility-based intervention activities implemented under NURHI-1 in Ilorin and Kaduna.

**Methods** A qualitative study that used in-depth interviews was conducted with 31 service providers purposively selected from 10 of the NURHI-1 intervention facilities in Ilorin and six in Kaduna. Interviews were digitally recorded and transcripts uploaded into ATLAS.ti for analysis. Structured observations to document renovations implemented during the NURHI-1 interventions were also conducted in the health facilities.

**Results** Family planning (FP) awareness creation within the facilities and integration of FP into existing maternal and child health and HIV services, were sustained in both cities. The majority of the equipment supplied as part of the NURHI 72-hour clinic makeover were still functional in both cities. Respondents in both cities reported that FP awareness and demand were sustained. On the whole, challenges with sustaining activities were reported more among respondents in Ilorin than Kaduna. In Ilorin, NURHI outreach activities and trainings, had discontinued while in Kaduna, they were no longer being implemented to the same degree as occurred during NURHI-1. Inadequate funds was a major reason for discontinued activities in both cities while integration of FP into existing services enhanced sustainability.

**Conclusions** Many activities were not sustained in Ilorin compared with Kaduna although FP awareness and demand remained high in both cities. Integration of FP into existing services promoted sustainability in Ilorin and Kaduna. A gradual closeout of donor projects with concomitant input from government and indigenous institutions could be useful in sustaining donor activities.

## INTRODUCTION

Consistent use of modern family planning (FP) methods has been found to have health,

**Strengths and limitations of this study**

► The study was led by investigators who have experience with designing and implementing qualitative studies and analysing qualitative data.

► Our choice of qualitative methods provided deeper insights into factors that promote or hinder sustainability of donor-funded programmes.

► The study used qualitative research methods, which limits generalisation of the findings. However, a range of participants who were knowledgeable in the issues under study were interviewed thus ensuring that findings are robust and credible.

► Participants from the Nigerian Urban Reproductive Health Initiative second phase (NURHI-2) sites could have experienced some difficulty in differentiating between NURHI-1 and NURHI-2 interventions. To minimise the likelihood of incorrect reporting, we clarified this difference to participants in the NURHI-2 site and emphasised that the questions were based on NURHI-1 interventions.

social and economic benefits for mothers, children and society at large.[1–7] FP benefits include reduced risk of maternal mortality and morbidity, improved infant and child health outcomes, and improved health and well-being of families.[1–5] FP has been identified as a key intervention towards meeting virtually all of the sustainable development goals.[6] Despite these clear benefits, FP use remains low in many developing countries especially in sub-Saharan Africa.[1 8]

Tsui and colleagues examined data on modern (modern contraceptive methods included the following: barrier methods (male and female condom, foam, gel), injectables, oral contraceptive pills, implants, intra-uterine devices) contraceptive use by married women ages 15–49 across four time periods and demonstrated that modern

contraceptive prevalence rates were consistently lower in Africa compared with other world regions.[9] In Nigeria, data from the 2013 Nigeria Demographic and Health Survey showed that 11.1% of all women aged 15–49 were currently using a modern contraceptive method.[10]

## Sustainability and factors influencing sustainability

Several initiatives and programmes have been implemented to improve uptake of FP in developing countries.[11] Many of these programmes were donor-funded and resulted in varied successes in terms of increased FP attitudes, knowledge, and uptake. The extent to which these successes have been sustained following project closeout remains debatable. The concept of sustainability, especially with respect to public health interventions, has therefore received a lot of attention. Various authors have defined what constitutes sustainability of health interventions and further described the factors that promote sustainability. Bamberger and Cheema defined sustainability of health interventions as, 'the extent to which an evidence-based intervention can deliver its intended benefits over an extended period of time after external support from the donor agency is terminated'.[12] Abrams and colleagues described it simply as, 'whether or not something continues to work over time'.[13] Shediac-Rizkallah and Bone described three measures of sustainability: (i) continued health benefits for individuals after the initial programme funding ends, (ii) continuation of programme activities within an organisation and (iii) continued capacity of a community to develop and deliver health promotion programmes.[14] A fourth measure of sustainability which refers to efforts undertaken to maintain the ideas, beliefs, principles or values underlying a particular programme has also been described by Weiss and colleagues.[15]

Shediac-Rizkallah and Bone developed a framework which highlights three important categories for sustainability: (i) project-related factors (ie, factors relating to the design and implementation of the project such as the resources available to the project and the project activities); (ii) factors within the organisation implementing the project; and (iii) factors within the broader community environment. Shelton and colleagues described an integrated sustainability framework which consists of multilevel factors that promote sustainability across multiple settings. These factors include (i) outer contextual factors, (ii) inner contextual or organisational factors, (iii) processes within the organisation, (iv) characteristics of the intervention and (v) characteristics of the project implementer.[16] Other factors that have been identified as essential for achieving sustainability include the use of a strategic approach for the intervention, adequate funds, community ownership and mobilisation and working with existing community resources (such as manpower, social institutions and sound infrastructure).[14 17–20] In an attempt to ensure sustainability of interventions, projects often incorporate a combination of these factors into their intervention package.

This study examines various components of programme sustainability with a focus on the sustainability of FP programming service delivery after donor funding ends.

## Description of the Nigerian Urban Reproductive Health Initiative

The Urban Reproductive Health Initiative, funded by the Bill and Melinda Gates Foundation, was inaugurated in 2009 with the broad goal of addressing and meeting the FP needs of the urban poor in four countries: India, Kenya, Nigeria and Senegal.[21 22] In Nigeria, the first phase of the Nigerian Urban Reproductive Health Initiative (NURHI) was implemented between 2009 and early 2015 in the Federal Capital Territory, Ibadan, Ilorin, Kaduna, Benin and Zaria. The overall goal of the NURHI was to promote innovative FP delivery approaches and increase modern contraceptive use in the six cities, with a focus on the urban poor. In 2015, the second phase of NURHI (NURHI-2) commenced with operations in three states: Lagos, Kaduna and Oyo. NURHI-2 expanded its focus beyond the initial urban approach to include state-level interventions in both rural and urban areas. NURHI-2 incorporated approaches to support sustainability of project activities and ultimately FP uptake in its intervention sites.

The NURHI intervention used a three-tiered model comprising advocacy, demand generation and service delivery to achieve its goals.[23] The focus of this paper is on the service delivery components of the NURHI programme. The NURHI service delivery approach included the following activities:

1. Health system strengthening such as the NURHI 72-hour makeover, community outreach, integration of FP services within existing maternal, neonatal and child health (MNCH) and HIV services; improving the Commodities Logistics Management System and setting up a referral system.
2. Quality improvement: this was achieved through competency-based training, integrated supportive supervision and on-the-job training, site orientation for NURHI high volume sites and distance education.
3. Promotion of public private partnerships to improve referrals across sectors and increase access to FP services.

The Measurement, Learning & Evaluation (MLE) project undertook the impact evaluation of the NURHI programme; this included baseline, midterm, and endline household and facility-based surveys.[24–26] Post immediate impact effects, it is important to assess the sustainability of the NURHI interventions when funding ends. This informs whether immediate gains are maintained and whether there are longer-term system-level and program-level benefits of the FP programme. It is also important to highlight areas that might require urgent interventions to maintain the achievements.

The focus of this paper is to describe the extent to which NURHI interventions were sustained and factors influencing sustainability in a city where NURHI activities

came to a close in 2015 (Ilorin) and a city where activities were modified and continued as part of NURHI-2 (Kaduna). Two measures of sustainability: (i) the extent to which the project activities continued in the target health facilities following closeout and (ii) perspectives of facility staff about the continuation of immediate effects of the intervention on the target population after project closeout were assessed. Factors promoting sustainability of the intervention activities were also explored.

Research questions:

1. Were there any changes in facility-level intervention activities after the NURHI-1 project closed?
2. Were there changes in immediate outcomes—attitudes and demand for FP after the project closed?

## METHODS

The NURHI sustainability study had two components: a quantitative component[27 28]; and a qualitative component from which data for the current paper is obtained. The qualitative phase of the NURHI sustainability study was led by researchers at the Centre for Population and Reproductive Health, College of Medicine, University of Ibadan, Nigeria with technical support from the MLE project at the Carolina Population Center in Chapel Hill, North Carolina, USA. Information was obtained from participants in Ilorin, Kwara state and Kaduna city, Kaduna state.

For the qualitative study, we undertook key informant interviews with government stakeholders, in-depth interviews (IDIs) with health facility staff, and focus group discussions with women from the study communities. This paper is based on the in-depth interview data from facility staff as the objective is to assess sustainability of facility-based interventions. In addition to the interviews, research staff also documented the availability and state of interventions implemented during the NURHI-1 programme using a structured checklist. These data collection methods were selected to enable investigators to obtain in-depth information from a range of facility stakeholders about the NURHI-1 interventions and underlying factors which facilitated or undermined sustainability.

### Study participants

Selection of the facilities for the qualitative study was informed by data obtained from the quantitative phase. In total, NURHI worked in 21 facilities in Ilorin and 23 facilities in Kaduna under NURHI-1. Quantitative data collected from facilities in 2017 were analysed to ensure that a mix of facilities categorised as high and low sustaining were selected. Factors taken into consideration included variables that would have been influenced by NURHI programming such as availability of Information, Education and Comunication (IEC) materials, training of staff within the last year, contraceptive method availability, existence of an outreach programme, and existence of a quality assurance committee. Based

on this process, IDIs were undertaken between June and July 2018 with facility staff (FP service providers) in 10 facilities in Ilorin (five that had positive indication of sustainability and five with worse indicators related to sustainability) and six in Kaduna (three with better and three with worse indicators). A total of 31 health facility staff (20 in Ilorin and 11 in Kaduna) were interviewed including the medical director or designate and a staff directly involved in providing FP services in each of the selected facilities. The study oversampled FP providers from Ilorin in order to get a more nuanced perspective of the impact of NURHI-1's closure. Only staff who had been working in the facility prior to commencement of the NURHI-1 intervention were eligible to participate. The total number of participants interviewed was deemed adequate to meet the study objectives, however, there was provision to recruit additional interviewees until saturation was attained.[29]

### Study instruments

The in-depth interview guide and health facility checklist were developed by a consultative process and all researchers agreed on the final version. The instruments were translated into Yoruba and Hausa, the predominant indigenous languages in the study cities, and back-translated to ensure the original meanings were retained.[30] The instruments were pretested among a comparable group of respondents and revised as required. A 3-day training was conducted for all research staff who conducted the interviews. The training comprised didactic lectures, case studies, role plays as well as practical field visits and interviews followed by presentations and feedback. The training was interactive and provided data collectors and the study investigators the opportunity to edit the instruments where necessary and discuss city specific issues which could arise and how to resolve these. In addition, the instruments were further reviewed following the training to ensure that locally acceptable terms were incorporated as required. The interviews were conducted by a team comprising an interviewer and a notetaker and each interview lasted approximately 45 min to more than an hour. The interviews were digitally recorded and the notetaker documented notes as well as non-verbal cues.

### Analysis

Data analysis was guided by a content analysis approach[31] and conducted by a team led by a qualitative researcher from MLE and an in-country researcher. Recorded interviews and discussions were translated and transcribed verbatim. Random checks of up to 10% of all transcripts against the recordings were conducted by the analysis team leads to ensure that the interviews were accurately transcribed. The transcripts were then uploaded into ATLAS.ti software (Scientific Software, Berlin; V.7) and coded.

Codes were developed using both deductive and inductive approaches and codebook development was informed by guidelines on 'Codebook Development for

Team-Based Qualitative Analysis' by McQueen *et al.*[32] This improved inter-coder agreement and facilitated coding and data analysis by the MLE and in-country team. The data collection team leads and interviewers were also involved in the process of code development and provided additional clarification as required.

Coding of the transcripts was conducted by a coding team comprised of a team lead and six coders. When discrepancies were noted, the coding team discussed these and the code to be applied was agreed on. The team lead independently conducted spot checks on randomly selected transcripts during the coding process. Coding continued until appropriate codes were applied to all segments of the transcripts.[29] For the current paper, the authors read the coded transcripts again and developed additional sub-codes where required. Codes were categorised into themes for each of the objectives and cross-case analysis performed using ATLAS.ti. (Additional details on data analysis in online supplemental appendix 1).

### Patient or public involvement
Research assistants were recruited from the general public and participated in the data collection. Patients were not involved in the study.

## RESULTS
### Characteristics of respondents interviewed in Ilorin and Kaduna
Thirty-one participants (20 FP providers in Ilorin and 11 in Kaduna) provided consent and were interviewed. One participant who initially agreed to take part in the study in Kaduna later declined for personal reasons.

Respondents' views on the sustainability of NURHI-1 activities and the factors promoting sustainability are presented in the following sub-sections:
1. Sustainability of NURHI-1 activities in Ilorin and Kaduna.
2. Sustainability of immediate outcomes, that is, improved FP awareness, perceptions and demand in Ilorin and Kaduna.
3. Factors promoting sustainability of the NURHI-1 intervention activities and demand for FP.

### Sustainability of NURHI-1 activities in Ilorin and Kaduna
Health facility staff in Ilorin and Kaduna were interviewed for their views about sustainability of the different facility-level NURHI-1 activities including advocacy and community mobilisation, service delivery and interventions to ensure quality services provision. Respondents in both cities, revealed that activities were either ongoing or had been discontinued. For ongoing activities, a few were being conducted to a comparable degree as occurred during the NURHI-1 intervention period while others were 'struggling', that is, they were no longer being carried out to the same extent as occurred while NURHI-1 was ongoing. These views cut across all categories of the NURHI interventions and were expressed by more respondents in Ilorin than in Kaduna.

### Advocacy and community mobilisation activities in Ilorin and Kaduna
Generally, the interviewees reported that the community-level advocacy and community mobilisation activities undertaken by members of the NURHI Advocacy Core Group were no longer being implemented to the same extent as during NURHI-1. The decline in implementation of these activities was predominately reported by participants in Ilorin, where NURHI-1 had closed out. Notably, some participants in Kaduna (where activities under NURHI-2 were still ongoing), also mentioned a decline in community-level advocacy and mobilisation which might reflect the focus under NURHI-2 on state-level activities and not just targeting of Kaduna city.

An FP provider in Ilorin described the state of things since the closeout of NURHI-1:

> … you know when NURHI was here, they helped us with many things. They had social mobilisers. These social mobilisers, they gave them token money for them to go out, to go inside the town, call people to come and do family planning. We had the ACG [Advocacy Core Group], they gave us money to do everything; but now, (there is) nobody to finance us.

Another FP provider in Ilorin remarked,

> We don't depend on … mobilization [for awareness generation] again; people are just coming in [for services]. Unlike before, when they [NURHI] were on ground, the mobilizers will go around the town, they invite people, they give them cards to come to this facility; but now that NURHI has left—nothing like that again.

Although most of the FP providers interviewed in Kaduna mentioned that community mobilisation and outreach were still ongoing, a few FP providers remarked that there had been a decline in these activities. For example, an FP provider in Kaduna explained,

> … the outreach; mostly it is done in the facility. Which is rather 'in-reach' to me, but then, when NURHI phase 1 was on, they [community mobilizers] do go into the community and they look for a strategic place, within the community and bring in people.

### Sustainability of service delivery interventions in Ilorin and Kaduna
Responses on the sustainability of the integration of FP into existing MNCH and HIV services were similar in both Ilorin and Kaduna. Respondents in both cities remarked that the integration had been sustained and enjoyed support from staff in these other units within the health facility. A respondent in Kaduna affirmed,

> We've integrated it [family planning] with the lab and [services for] people living with HIV. There is a training we undergo. That's why the PMTCT unit is

just here and the FP [unit] is here. So that it will not be far, so any person coming for PMTCT, if she's a married woman, they will advise her to come to the FP unit, so she'll come for the FP and we will counsel her. And there are some women that come here purposely for FP, then we send them to the lab for HIV test.

Although FP had been well integrated into existing service units, stock-out of FP commodities and consumables was a challenge in the post-NURHI-1 period. This was reported more often among respondents in Ilorin than in Kaduna. A provider in Ilorin stated,

Earlier this year, even at present, if I show you our cupboard, there is no Implanon because the one that we have has finished, so we don't have. And I requested, I made requisition from my supervisor. She told me that it's not available and the IUCD too is not available.

In Kaduna, an FP provider corroborated the issue of stock-out of commodities and efforts made to ameliorate this at the facility level which included reaching out to other facilities for commodities, 'The facility cannot do anything. We used to go to Kaduna south, sometimes their supply comes from the federal level so we can get what we want exactly. Sometimes when we request for hundred, we get fifty so that is why we get the stock-out'.

We noted that although some respondents in both cities initially said they did not experience commodity stock-out, after probing for details of availability of the different FP commodities, it was discovered that virtually all government facilities, in both Ilorin and Kaduna, had experienced stock-out of at least one of the methods since NURHI-1 came to a close. Thus, providers had sometimes been constrained to offer clients an alternate method if the preferred method was unavailable. Excerpts from an interview with an FP provider in Ilorin illustrating this is outlined below:

Respondent: Since I've been here, I have not experienced any stock-out, since February. Except the month that ehnm … 'Depo' injection [Depo Provera], that injectable was very scarce.

Interviewer: …, so during that period when the Depo wasn't available, what did you do for clients that came to seek that particular method?

Respondent: so, hmm, we still advised them [to take] Noristerat … we told them that if 'Depo' is available we will give them … (FP Provider, Ilorin)

A provider in Kaduna also described the situation thus:

The problem of stock-out we normally have is from [inadequate supply in] the whole state and [occurs] once in a while, but not always. There was a time Implanon was not available and they will say they have not collected delivery from the Federal [level]. So, it will be a general issue. … And the microgynon, we

used to have stock-out of microgynon,—sometimes, not always.

FP providers in private and tertiary health facilities in both cities did not report having problems with stock-out of commodities as they obtained commodities from other sources and paid for these. An FP provider in a tertiary facility in Ilorin explained, 'We don't have stock-out. We have all the methods on ground now. We don't have stock-out and they [clients] can come for any method they like. In this facility, we offer all methods'.

A provider in a private facility in Kaduna remarked, '… once it is end of the month, I prepare a report so that there wouldn't be any stock-out'.

The problem of stock-out of FP consumables was quite common both in Ilorin and Kaduna. Many of the respondents had experienced stock-out of consumables such as cotton wool, gloves, needles and syringes, and bleach. An FP provider in Kaduna specifically mentioned the challenges with availability of consumables experienced since NURHI-1 transitioned to NURHI-2.

You know now, they don't concentrate more in the urban, they are now in the rural areas, so like now the consumables they used to give us has reduced. [It's] not like before, not like NURHI-1 and that is understood because they are facing (focusing on) the rural areas, the grassroots.

Stock-out of consumables was not a problem in private facilities in either of the two cities as these facilities billed patients for services including FP services. They could thus afford to procure consumables privately and not depend on government supply or supply by non-governmental organisations (NGOs).

### Sustainability of quality FP service provision in Ilorin and Kaduna

Interventions to ensure quality FP service provision such as competency-based training and on-the-job training and site orientation for NURHI high volume sites had been significantly affected by the NURHI-1 closeout in both study cities. Facility staff in Ilorin reported that the NURHI-type of trainings had become non-existent. A provider in Ilorin stated,

At this time I want to implore the NGOs on the training and retraining of staff on FP. There are many staff that have interest, when there is no training for them, there is nothing we can do. Training and retraining are very, very necessary.

A similar observation was made by FP providers in Kaduna; 'NURHI has not been training our staff again, that's one of the changes'.

In addition to the fact that NURHI trainings had been discontinued, providers mentioned that a number of the facility staff who had been trained had either retired, were approaching retirement or had relocated to other health facilities or states where they would be better remunerated. This had depleted the pool of trained FP providers.

Another important NURHI-1 intervention undertaken to strengthen the health system was the 72-hour makeover which involved facility renovation and supply of equipment. The majority of the activities implemented under the 72-hour makeover renovations had been sustained in both cities. The equipment supplied were still largely available and functional. Examples of quotes from FP providers in Ilorin and Kaduna illustrating the current state of the renovations carried out are below:

> The equipment are still intact and all other things that they put in place like; the scale, equipment to check their BP. (FP provider, Ilorin)

> We are maintaining what they gave us to work with; like the instruments, the chairs outside. If they break, we need to repair them to make everything go well. (FP provider, Kaduna)

A provider in Kaduna mentioned that he spearheaded maintenance activities within his facility. He commented, 'As you can see, it's not long they repaired this place. I actually did the repairs. From my micro plan, I did the repairs'.

Neither respondents in Ilorin nor those in Kaduna had witnessed a major large-scale renovation in their facilities since the end of NURHI-1, although minor repairs had been carried out. A provider in Kaduna commented,

> We have not been seeing some things again from them (NURHI). Like all these chairs are from NURHI and the tables and some of the things are from NURHI. In fact, like the chairs, the tables and all these things in the facility—in this unit are from NURHI, but now we no longer see anything again.

As part of the study, interviewers observed the premises of the facilities for evidence of sustainability of the 72-hour health facility renovations. These findings largely corroborated the reports of the interviewees. A higher proportion of respondents in Kaduna than in Ilorin affirmed that their facilities had benefited from the NURHI 72-hour makeover, however, the makeover plaques were seen in a higher proportion of facilities in Ilorin than in Kaduna. The environment around all the health facilities surveyed in Ilorin and most of facilities in Kaduna were clean and well-kept at the time of the visit. Many facilities in Ilorin and Kaduna had spaces allocated to FP which were conducive and offered privacy for clients. All the facilities in Ilorin and in Kaduna had FP IEC materials displayed and NURHI-IEC materials were seen in all but one facility in Kaduna (table 1).

Overall, lack of funds was a major factor cited as responsible for the cessation or decline in activities in Ilorin and Kaduna. While respondents in Ilorin attributed this lack of funds to the closeout of the NURHI-1 intervention, respondents in Kaduna often attributed this to the modification of the focus of the NURHI-2 interventions which expanded to include rural communities beyond Kaduna city. Hence, some respondents in Kaduna were

**Table 1** Sustainability of facility renovations as observed by interviewers

| | Ilorin | Kaduna |
|---|---|---|
| | n=10 | n=6 |
| | n (%) | n (%) |
| NURHI 72-hour makeover conducted | 6 (60%) | 5 (83.3) |
| NURHI 72-hour makeover plaque sighted | 4 (66.7) | 3 (60.0) |
| Clean and well-kept FP environment | 10 (100.0) | 5 (83.3) |
| Conducive space that affords auditory and visual privacy allocated to FP | 8 (80.0) | 3 (50.0) |
| FP IEC materials seen | 10 (100.0) | 6 (100.0) |
| FP IEC materials produced by NURHI seen | 10 (100.0) | 5 (83.3) |
| FP IEC materials targeting adolescents and youth seen | 1 (10.0) | 1 (16.7) |
| FP IEC materials produced by NURHI targeting adolescents and youth seen | 1 (100.0) | 0 (0) |

FP, family planning; IEC, Information, Education and Communication; NURHI, Nigerian Urban Reproductive Health Initiative.

no longer experiencing the effect of the NURHI-2 activities compared with the situation under NURHI-1.

### Sustainability of immediate outcomes of NURHI-1 interventions in Ilorin and Kaduna

Respondents in both cities affirmed that the NURHI-1 intervention had improved FP awareness, perceptions and demand for commodities in their catchment areas. They further acknowledged that although NURHI activities had declined, the effects of these activities were still being felt. An FP provider in Ilorin remarked,

> Before, there was not much of awareness—before NURHI came. People didn't know much about family planning. Even people that knew, they just knew; maybe about the pills and injectables. But now that people are aware that there is long acting (contraceptive), they tend to go for the long acting rather than the short (acting) ones.

She further added, 'We have people that have busy schedules, like those people working in the bank, you don't see them before, but now you see them coming for those long acting [methods], they come for IUCD or Implanon' (FP provider, Ilorin).

Another provider in Ilorin affirmed that demand for FP services had increased as a result of the NURHI-1 intervention.

> It has increased because there is more awareness now, people get to know more about family planning and during the antenatal clinic, they are getting

information. Even some, before they deliver, you will see them coming for counseling that immediately they deliver or as soon as they deliver, they want to come for family planning. So they are more knowledgeable now and this makes them to be coming. (FP provider, Ilorin)

Similar views were expressed by FP providers in Kaduna, who admitted that demand for FP, especially Long-acting Reversible Contraceptives (LARCs), had increased, as a result of the NURHI community awareness outreaches. One of the FP providers interviewed had this to say:

More people are asking for it, you know it's a long term method and people are thinking for them to go, comeback after 2 months, still go and come back after 3 months, they just feel the long acting method is the best. They tell you, 'I prefer the long acting method because once I just put it on, I'll just go on my own, I'll forget I'm putting it, no need of me telling myself I have to go back in 2–3 months, so it's preferable I go for the long term method'. (FP provider, Kaduna)

The NURHI 'Get It Together' edutainment radio programme was also noted to have contributed to improved awareness about FP and subsequently demand for FP commodities. This finding was reported by respondents in both Ilorin and Kaduna although the radio programme had been discontinued in Ilorin following closeout of NURHI-1.

### Factors promoting sustainability of NURHI-1 activities

Generally, many of the activities were no longer being implemented to the same extent as occurred during the NURIH-1 period in both Ilorin and Kaduna. Activities were however better sustained in Kaduna than in Ilorin. Respondents in both cities provided insights into factors that had promoted sustainability of some activities following NURHI-1 closeout in Ilorin and transition to NURHI-2 in Kaduna. These are presented under the following key themes that emerged: project-related factors, factors within the health facility and factors within the larger community.

### Project-related factors promoting sustainability of NURHI activities in Ilorin and Kaduna

A key project-related factor that promoted sustainability was the integration of FP awareness generation and service provision into existing activities and services within the health facilities. This was reported by providers in both Ilorin and Kaduna. In addition, the 72-hour makeover gave the facilities a face-lift and attracted clients to the facility. A respondent in Ilorin when asked about the immediate effect of the makeover remarked, 'people are trooping in; we have inflow, high inflow of ehn … client to this facility because through the light [sign] that was placed outside, it attracted people, even those that are bringing their children for treatment here in the hospital' (FP provider, Ilorin).

A provider interviewed in Ilorin stated that staff in the health education unit in the facility had incorporated FP information into routine immunisation and antenatal care clinics thereby facilitating continuation of FP awareness creation activities. She explained:

We normally have our National Programme on Immunization (NPI) sessions—Mondays and Wednesdays; then they talk to them about family planning during that period—Mondays, Wednesdays and Fridays. Especially for mothers who have just given birth and came for their routine immunization for their children. So we talk to them about family planning in this facility.

Similar accounts were shared by FP providers in Kaduna. In addition, some facilities were said to have involved Voluntary Community Mobilizers (VCMs) and Traditional Birth Attendants (TBAs) in creating awareness of FP within the community.

Yes, the VCM are playing their own role, the TBA's too are playing their own role in family planning by educating women out there at the community level. Then during routine immunization where we gather many of the women that bring their children for immunization, we use that opportunity too because at that time we can really get some of them. Then during antenatal again, we educate them about choosing a method of their own choice. So there is great improvement. (FP provider, Kaduna)

### Health facility-related factors promoting sustainability of NURHI activities in Ilorin and Kaduna

Some respondents cited motivation of health facility staff as a factor promoting sustainability of NURHI activities in both cities. Staff demonstrated their motivation in various ways for example, using personal allowances to purchase minor equipment and stepping down the FP training they had received from NURHI to other staff. An FP provider in Kaduna mentioned that he and other colleagues sometimes used their transportation stipends to purchase minor equipment. He explained, 'I was able to get these [pointing to a gallipot and some forceps]—some gallipots, forceps, from what they give us (our transport stipends)'.

Another FP provider in Kaduna explained that on account of the increased demand for FP, management of the facility planned to increase the number of staff in the FP unit and provide training for them to enable them to provide quality FP services to the increasing number of clients. He explained thus:

Because of the increase in number of patients, we plan to put more staff in the unit and then to do step down training and then teach them the procedures so that they can cope with the challenge and increased numbers of patients.

In contrast, direct efforts by the facility management to increase the staff strength was not mentioned by the respondents in Ilorin.

## Factors within the larger community promoting sustainability of NURHI activities in Ilorin and Kaduna

Two main factors within the larger community—the presence of other NGOs and to some extent, government efforts, were said to have contributed to sustainability of NURHI activities. Respondents in Ilorin and in Kaduna (in facilities that were not included in the NURHI-2 intervention or where providers stated that the impact of NURHI activities had waned under NURHI-2) mentioned that other NGOs working to improve FP in their states had promoted sustainability of NURHI interventions. Some of these NGOs included Pathfinder International, Society for Family Health and Marie Stopes International. These NGOs provided funds, FP training, commodities and consumables, and they were also involved in awareness generation activities. A special government-initiated intervention—Saving One Million Lives Initiative—was said to have contributed towards sustainability of some NURHI activities in Ilorin and Kaduna.

Providers in Ilorin mentioned that outreach, community mobilisation and awareness creation activities sponsored by other NGOs within the state had contributed towards sustainability of some of the activities NURHI had implemented and had also sustained FP demand generation. An FP provider in Ilorin explained,

> We used to organize outreaches, we send some people out to go and tell people, to go from house to-house …, just to encourage them to come out for family planning. … especially Marie Stopes—you know Marie Stopes has taken over after NURHI left.

Another FP provider in Ilorin expressed concern about the initial decline in clientele experienced when NURHI-1 came to an end. She however mentioned that the number of clients had again increased when another NGO commenced an FP intervention project in the city. She explained thus,

> Although when they (NURHI) were here—the population of clients then was higher. The moment they (NURHI) left, things fell apart. But now, when another NGO—Marie Stopes came, then we are improving again. Because, before, in this clinic we had 500, 600 clients per month, this declined to 100—plus, but now we have improved to 200, 300 (clients).

Additional ways through which other NGOS had contributed to the continuation of some of the NURHI activities were through training of healthcare workers in the provision of FP services and provision of IEC materials. NGOs had also been instrumental towards mitigating stock-out of commodities and consumables in Ilorin and in some facilities in Kaduna.

> Interviewer: Marie Stopes that you mentioned, how did they contribute to family planning service delivery in this facility?
>
> Respondent: Thank you. Through their trainings, then those that they trained, trained others too, then the knowledge people gained was a sort of contribution to family planning especially on these new methods. (FP Provider, Ilorin)

In Ilorin, the College of Nursing and Midwifery was said to be organising a training in FP use which all nurses working in FP units were encouraged to attend in order to acquire necessary skills. The training was however not free and would only hold if a minimum number of participants were registered. An FP provider provided the following information:

> So that one (training) is for (organised by) the school of—, College of Nursing and Midwifery. They are the ones organizing that family planning stuff. They should normally have 50 students at a time; so if they don't have this number, it may delay the training. To strengthen family planning, four people went last year but this year we are planning for them to go so that every staff will have knowledge that they need about family planning. So that's the plan we are making here.

A special government-initiated intervention—Saving One Million Lives Initiative—was said to have contributed towards sustainability of some NURHI activities in Ilorin and Kaduna. Responses on government efforts to sustain the NURHI activities and provide funding for FP commodities and consumables and for minor repairs within the health facilities differed. While some respondents acknowledged government efforts, others were of the opinion that the government had not done enough but that NGOs were the main supporters of FP services. The differing opinions were expressed by respondents in both Ilorin and Kaduna. A provider in Kaduna commented: 'the state government is actually trying. They make sure that there is no stock-out'.

Another FP provider in Kaduna however held a contrary view and when asked if the state government or other organisation had assisted with minimising stock-out in the last 5 years, he responded, 'not really, it's just partners like Marie Stopes in collaboration with the state, then this NURHI, Planned Parenthood Federation of Nigeria (PPFN)'.

## DISCUSSION

The paper describes the extent to which the activities of NURHI-1 have been sustained in Ilorin 2 years after closeout of the initiative and in Kaduna where the programme continued but with a slightly different scope and focus. Our findings revealed that in Ilorin, integration of FP awareness creation activities and FP services into existing MCH services and the renovations and equipment purchased as part of the 72-hour clinic makeover had been sustained. Community outreach and availability of commodities and consumables were not sustained to the same degree as they had been during the NURHI-1 period. The NURHI-facilitated FP

training had been discontinued. The situation in Kaduna, which is still benefiting from NURHI-2 support, was slightly better than in Ilorin, although respondents noted that interventions under NURHI-2 were not as robust as they had been during the NURHI-1 implementation. The NURHI-1 midterm and immediate closeout evaluations reported much higher degrees of sustainability compared with findings in the current study conducted about 3 years after NURHI-1 closeout.[24 25] Our findings thus indicate a progressive decline in project activities as the duration from NURHI-1 closeout to time of evaluation increases.

Common reasons given by respondents in both cities for failure to sustain NURHI-1 activities were inadequate funds and problems with the commodity supply chain, which could also be linked to funding. Other authors have underscored the importance of funding in sustaining health and other programme interventions.[33–37] Shediac-Rizkallah and Bone emphasise the need for projects to have strategies to facilitate gradual financial self-sufficiency in order to promote sustainability.[35] The transition from NURHI-1 to NURHI-2 in selected cities including Kaduna could be an attempt to facilitate this gradual financial self-sufficiency, however, our findings suggest that this transition might not achieve this effect as many NURHI-1 activities were declining in Kaduna as was also reported in Ilorin.

Stock-out of FP commodities and consumables was reported in both study cities in spite of the robust advocacy component of NURHI-1, which contributed to the creation of government guidelines and strategies such as the Nigeria Family Planning Blueprint (Scale-Up Plan) 2014,[38] a dedicated FP budget line, and increased fund allocation to FP by the Nigerian government. With these government interventions in place, it is expected that there would have been adequate funding to continue NURHI project activities. This, however, was not the case and a possible explanation might be that although these policies and guidelines had been approved, the systems to ensure prompt implementation at all levels, especially down to the health facility levels, were still nascent and yet to be fully operational.

Findings on sustainability of immediate outcomes of NURHI-1 suggested that FP awareness and demand remained high even after closeout in Ilorin and in the continuation phase in Kaduna. That said, concerns about declines in FP demand were raised by some respondents in Ilorin. The sustained FP awareness and demand were attributed to the widespread community advocacy and mobilisation, awareness creation within the facilities as well as availability of a range of FP methods during NURHI-1. Reviews of literature on programme sustainability have reported that community engagement, which was an important aspect of NURHI-1, promotes programme sustainability by changing social norms in support of the programme.[18] It is important to note that the recurrent stock-out of commodities and consumables experienced in Ilorin and to some extent in Kaduna could potentially erode these immediate gains. It is thus important for the issue of stock-out to be urgently addressed and for measures to be put in place in the NURHI-2 sites to prevent recurrence of stock-out.

Factors which promoted sustainability from the respondents' views were integration of FP awareness generation and services into existing services and programmes, the 72-hour clinic makeover that increased patronage, and activities of other NGOs and some special government intervention programmes. NURHI-1 activities that leveraged on existing structures such as integration of FP service provision within existing MNCH and HIV services were sustained. Integration of project activities within existing programmes and services has been identified as an important factor that promotes sustainability.[35] Integrating FP into other existing health services is one of the key strategies being explored to improve FP availability and demand in Nigeria and this could also have contributed to sustainability of this intervention beyond the NURHI-1 period.[38] The 72-hour clinic makeover included provision of equipment which were still functional at the time of the study. Sustainability of this intervention in the short-term could be attributed to the fact that if properly used, the equipment would typically last for a while before needing to be replaced. The collaborative approach adopted for the 72-hour makeover involved prioritisation and costing of items that required replacement or repair by facility staff and key community members and use of direct labour for repairs. Although not mentioned by the respondents in either of the two cities, involving the staff in the process of renovation could have contributed to sustainability by promoting ownership and careful use of the equipment.[39]

Staff motivation also contributed to sustainability of some activities in both cities such as their efforts to step down the FP training received to newer staff. While the fidelity of this step down training might not be the same as the initial training provided, using a training-of-trainers model and providing training manuals can enhance sustainability and scale up of training interventions.[40 41] In Ilorin, a respondent mentioned that the Nursing and Midwifery College had an FP training targeting the nurses. This suggests that embedding the training component within an existing system or collaboration with existing professional organisations could be useful in sustaining manpower capacity building efforts of health projects. This further emphasises the importance of integrating project activities within existing systems and structures. The activities of other NGOs, which sustained some activities, cannot be said to promote sustainability in a real sense as the activities of the new NGO are also limited by the scope and timeline of the project. Similarly, government special programmes, if not integrated into existing programmes for continuity, are also limited in duration.

On the whole, challenges with sustainability of activities were reported more by respondents in Ilorin compared with their counterparts in Kaduna. This was probably because NURHI activities came to a close in 2015 in Ilorin (a NURHI-1 site), whereas activities were ongoing in a modified format in Kaduna as part of NURHI-2. A more gradual closeout of NGO-driven activities to allow the government to fully take over the programme, as suggested by a stakeholder in Kaduna, might have provided more time for the mechanisms for ensuring that the effects of the approved increment in government funding for FP are launched and felt at the

facility level; this might have prevented the decline in activities observed in Ilorin.

## LIMITATIONS

Our study had a few limitations. We acknowledge the limitations inherent in qualitative research methods and thus do not attempt to over-generalise our study findings. Furthermore, we did not define a hypothesis to test factors associated with sustainability. Our respondents in both cities however provided detailed responses describing their views about factors that promoted sustainability of the NURHI intervention in their locations. The study participants' views on sustainability could have been limited based on their job description and cadre within the health system. In addition, findings on the impact of the interventions on sustaining FP demand were based on the views of the providers which could have been over-estimated. We however noted that the providers' remarks did not only attest to sustained FP demand as some of them also mentioned that demand had reduced following closeout and only increased after another NGO commenced FP interventions. To improve the quality of our data, we selected a range of participants who were knowledgeable about the study questions and we compared findings across respondents within each city. We also included a structured observation to objectively assess sustainability of some of the NURHI interventions. Participants from the NURHI-2 sites could have experienced some difficulty in differentiating between NURHI-1 and NURHI-2 interventions. In order to minimise the likelihood of incorrect reporting, we clarified this difference to participants in the NURHI-2 site and emphasised that the questions were based on NURHI-1 interventions. We have reported comprehensive and in-depth findings obtained through rigorous qualitative methods and these provide useful insight into the extent to which the NURHI-1 activities were sustained.

## TRUSTWORTHINESS OF DATA

The design, data collection and analysis were informed by guidelines for ensuring trustworthiness of qualitative data described by Guba[42] and Shenton.[43] These guidelines mention four issues (credibility, transferability, dependability and confirmability) as key to achieving trustworthiness of qualitative data. The investigators made efforts to ensure the study design, data collection and analysis met these criteria (see details in online supplemental appendix 2).

## CONCLUSION

On the whole, results of this sustainability study revealed that while a few of the NURHI initiated interventions were sustained, many more had declined or discontinued, particularly in Ilorin where the programme ended. In addition, although most activities were not being carried out as before, their effects such as improved demand for FP commodities and improved perception about FP were reportedly still being felt. These findings affirm the importance of incorporating a range of strategies to promote sustainability of donor projects that aim to improve FP demand and uptake in our setting. Lack of funding for continuing project activities was a major deterrent to sustainability reported by our respondents, hence it is important for donor projects to develop a plan which facilitates gradual financial self-sufficiency of the project beneficiary so that activities can be sustained following project closeout. Selecting project activities which align with government plans and integrating these activities within existing services could also increase the likelihood of sustainability. Finally using a training model that not only improves the competence of the trainees but also builds their capacity to step down the training could be useful for promoting sustainability of similar donor-funded projects.

**Author affiliations**
¹Institute of Child Health, College of Medicine, University of Ibadan, Ibadan, Oyo State, Nigeria
²Center for Population and Reproductive Health, College of Medicine, University of Ibadan, Ibadan, Oyo State, Nigeria
³Carolina Population Center, University of North Carolina at Chapel Hill, Chapel Hill, North Carolina, USA
⁴Department of Maternal and Child Health, University of North Carolina at Chapel Hill, Chapel Hill, North Carolina, USA
⁵Department of Obstetrics and Gynaecology, College of Medicine, University of Ibadan, Ibadan, Oyo State, Nigeria

**Acknowledgements** We acknowledge all study participants and the research staff who worked on the project.

**Contributors** AOO was the in-country study supervisor for the study and contributed to the review of the protocol, editing and revision of the study instruments, training and supervision of field staff, data collection and analysis. CM was the qualitative project investigator and involved in development of the protocol and study instruments, training and supervision of field staff and data analysis. LC was a project investigator and involved in development of the protocol and study instruments and data analysis. IS was the study principal investigator and provided oversight for the entire project. She was also involved in development of the protocol and study instruments and review of the data. TB was the in-country field supervisor and contributed to the review of the protocol, editing and revision of the study instruments, training and supervision of field staff, data collection and review of the data. OO was the in-country principal investigator and provided in-country oversight for the project. He contributed to the review of the protocol, editing and revision of the study instruments, training of field staff supervision of data collection and review of the data. All authors contributed to the write-up and revision of the manuscript and approved the final version submitted for publication.

**Funding** This study was funded by a grant from the Bill & Melinda Gates Foundation to the Carolina Population Center at the University of North Carolina at Chapel Hill. This research also received support from the Population Research Infrastructure Program (P2C HD050924) awarded to the Carolina Population Center at The University of North Carolina at Chapel Hill by the Eunice Kennedy Shriver National Institute of Child Health and Human Development. The contents of this article are solely the responsibility of the authors and do not necessarily represent the official views of CPC or the Bill and Melinda Gates Foundation.

**Competing interests** None declared.

**Patient consent for publication** Not required.

**Ethics approval** Ethical approval for the study was obtained from the National Research Ethics Committee (protocol number: NHREC/01/01/2007-25/04/2017) and the Institutional Review Board at the University of North Carolina at Chapel Hill (study number: 17-1215). All participants provided verbal consent and they were assured that their responses were confidential and they could discontinue the interview at any point if they desired to do this. In addition, the interviews were all conducted in private locations where participants were free to express their opinions.

**Provenance and peer review** Not commissioned; externally peer reviewed.

**ORCID iD**
Adesola Oluwafunmilola Olumide http://orcid.org/0000-0003-4372-9822

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
