## [Reviewer comments · BMJ Open]

ARTICLE DETAILS

TITLE (PROVISIONAL)	Factors promoting sustainability of NURHI program activities in Ilorin and Kaduna, Nigeria: Findings from a qualitative study among health facility staff
AUTHORS	Olumide, Adesola; McGuire, Courtney; Calhoun, Lisa; Speizer, Ilene; Mumuni, Tolulope; Ojengbede, Oladosu

VERSION 1 – REVIEW

REVIEWER	Juliet Iwelunmor Saint Louis University, USA
REVIEW RETURNED	06-Nov-2019

GENERAL COMMENTS	Introduction: Strength: The background information on modern family planning methods are sufficient. The authors laid out their argument on the importance of family planning and why it matters for women in Nigeria. Description of the NURHI program was adequate particularly the different parts focused advocacy, demand generation and service delivery. Weakness: The portion beginning the "NURHI Sustainability Study" is limited in scope and intent. No clear definition of sustainability or why it matters, not theory used to guide how they defined sustainability, not clear whether they were interested in sustainability at the individual level (which would matter given the focus on modern family planning) or at the program or organizational level. If the authors are going to do a rigorous study on sustainability, look into existing systematic reviews and viewpoint papers on what is sustainability and why it matters. This lack of attention to the notion of sustainability weakened the overall paper. Methods: Use of qualitative methods to understand sustainability are adequate. However, even though they use constant comparison approach, a derivative of grounded theory, no theory on sustainability was derived. Results: Weakness: a qualitative paper, that used constant comparison, with results still focused on sustainability as the lead heading, highlights the weakness in the actual data analysis portion. The themes generated do not correspond to the respondent comments "nobody to give them stipend." If lack of funds or limited mobilization or outreach were the reasons for example, the advocacy or demand generation programs were not sustained, then highlight this, rather than having the results in the same manner as the description of the NURHI program. More data analysis is required given the authors use of the qualitative
---

	research and I highly recommend using a standard text or article on what qualitative data analysis entails with revisions. Overall, the results section were poorly written with respondents quotes that do not correspond to the themes generated or rather the NURHI program described again in the results. Discussion: This section started to describe sustainability. The authors should have started their paper with this definition as it would have helped to address some of their weakness. In fact, they should go back and use the papers on their reference list, from #28-33 and 35-38 as a guide for their revisions.
--	--

REVIEWER	Win Brown Bill & Melinda Gates Foundation Seattle, WA USA The funding for the sustainability study does come from the Bill & Melinda Gates Foundation.
REVIEW RETURNED	22-Nov-2019

GENERAL COMMENTS	I have heard about this study to detect whether NURHI had any sustained effects beyond the funding period(s), and believe that such findings are important for our field. This particular qualitative submission, however, does not sufficiently answer the question. What's needed is a more clearly designed way to establish a contrast between the responses from Ilorin (where NURHI activities were not continued in Phase II of the project), and the responses from Kaduna (where NURHI activities were continued in Phase II of the project). This contrast is the basis for detecting differences in responses, yet many of the qualitative observations - though informative - were not presented within a comparative "between-group" context, as you might expect in an evaluation framework. Moreover, the reader expects to see observations from a third city, Jos, where NURHI was never implemented. But while Jos is mentioned at the outset, it never appeared in the results. Such observations from a "control" site like Jos would add to the descriptive and explanatory findings. More attention to structuring the responses within a comparative framework would be useful. Otherwise, though unintentional, the reader is likely to interpret each of the qualitative findings as an independent observation, rather than an observation that is dependent on group membership (i.e. belonging to Ilorin, to Kaduna, or to Jos). At times it was not clear whether the respondents were qualified to speak of NURHI effects beyond the service provision & supply dimension of the project, because the understanding is that most respondents were service providers. Therefore it isn't clear whether their remarks about advocacy or demand effects were as valid as, for example, focus group discussions from the community might have been. A small note on testimonials: though the rigor and integrity paid to translation are commendable, often the testimonials were difficult to read and interpret. NURHI also significantly changed its focus from its initial funding phase to its follow-up funding phase. The project shifted from a focus on the "urban poor" among six cities, to a focus on both urban and rural populations (poor?) in 3 States. For sustainability evaluation, this shift is problematic, because one would assume that during the 2nd phase of the project the units of analysis are
---

harder to maintain, and establishing evaluative equivalence between the two phases becomes more difficult. How this affects the study design and the research questions is not clear.

Speaking of research question, the reader is faced with at least two ways to observe "sustainability." The first is whether implementation of NURHI led to a sustainable impact on FP outcomes. The second is whether any aspects of NURHI implementation - any of its activities - continued after the project was closed. The two are not the same, but both types of sustainability effects are found in the observations and in the results. Can these two different indicators of sustainability be differentiated more clearly? Perhaps, but they are certainly both important, and the paper reveals promising findings in both of these areas.

With these things in mind, what's missing is a clear hypothesis for how a major project like NURHI would "continue" (both in terms of impact and activities) after donor funding is terminated. Factors that contribute to continuation are discussed, but the discussion is not framed as a hypothesis that can be supported (or not) by the resulting qualitative findings.

For example, one hypothesis might be that when NURHI demonstrated treatment effects and efficiencies in activities that were priorities for the GON FMOH or MOH programs, they would be more likely to persist (in some form) beyond the life of the donor project. That indeed might be the case when the paper discussed the apparent sustainability of FP's integration with MNCH and HIV activities, because the GON has prioritized an integrated PHC approach nationwide, etc.

For the most part, "sustainability" of NURHI was not confirmed in many respondent testimonials because stipends were no longer available to have workers do their jobs. But this is obvious. What the reader wants to know is how to design a series of project-based interventions that contain specific mechanisms for maximizing the possibility that project activities and effects will be "taken-up" after the project ends.

What are these mechanisms? Training, for example, is often discussed as a priori possessing more of a "sustainability" dimension, because a project focused on training can be viewed as contributing to human capital development, and therefore to system strengthening.

In this context, the section on the NURHI 72-hour clinic makeover activity was very interesting. Was it a health system-strengthening activity? Was it sustainable? If no, why not? This activity is often cited as leading to substantial gains in modern contraceptive prevalence, but from this paper the reader is hoping to learn - more conclusively - whether the 72-hour clinic makeover was (1) designed with sustainability in mind; and (2) achieved sustainability. The reader is close to getting an answer, but not a complete one.

Finally, the authors present a good list of recommendations for sustainability. However, the list is not a logical and direct extension of the observations contained in the body of the paper. i.e. the list

	does not appear to be directly informed by and resulting from the way in which the qualitative findings were presented.
--	---

REVIEWER	Philip Anglewicz Johns Hopkins Bloomberg School of Public Health, USA
REVIEW RETURNED	27-Dec-2019

GENERAL COMMENTS	Overall, this research is interesting and fairly important, but there are some items that should be addressed before it is considered for publication. It would be useful to have more information about the sampling approach (pg 6). What is meant by "informed by data obtained from the quantitative phase"? Were all facilities from the first stage chosen, or only a subset? How were participants selected, and how was the number of participants determined? Since there were other qualitative approaches used by the study, why does this manuscript report only results from the in-depth interviews, leaving out the focus groups and KII? The results seem incomplete without the other qualitative data. It would be useful to have a more detailed description of the qualitative methods in the abstract. The manuscript would benefit from a thorough review of grammar, formatting, and spelling. For example, there are statements/phrases such as "over-generalization of the findings", "Demand for family planning services generated by NURI activities were...",
---

VERSION 1 – AUTHOR RESPONSE

Reviewer: 1

Introduction:

Strength: The background information on modern family planning methods are sufficient. The authors laid out their argument on the importance of family planning and why it matters for women in Nigeria. Description of the NURHI program was adequate particularly the different parts focused advocacy, demand generation and service delivery.

Weakness: The portion beginning the "NURHI Sustainability Study" is limited in scope and intent. No clear definition of sustainability or why it matters, not theory used to guide how they defined sustainability, not clear whether they were interested in sustainability at the individual level (which would matter given the focus on modern family planning) or at the program or organizational level. If the authors are going to do a rigorous study on sustainability, look into existing systematic reviews and viewpoint papers on what is sustainability and why it matters. This lack of attention to the notion of sustainability weakened the overall paper.

Response: Thank you for your very useful comments. We have addressed them and believe this has improved on the paper as a whole. The section we initially developed on sustainability had been taken out in order to keep within the word count specified. We have now included it in the Introduction and have further improved on it in line with all the suggestions provided. The section in the discussion has also been moved up and is now included in the Introduction. We have included a section on the sustainability framework as well.

Methods: Use of qualitative methods to understand sustainability are adequate. However, even though they use constant comparison approach, a derivative of grounded theory, no theory on sustainability was derived.

Response: Thank you for this comment. We did not derive any new theory on sustainability. We approached the data analysis from the point that our respondents could offer newer perspectives on sustainability and the factors that promote sustainability which could result in a new theory or the modification of an existing theory. Our participants' reports largely aligned with previously described frameworks with emphasis on funding as a cause of lack of sustainability.

Results:

Weakness: a qualitative paper, that used constant comparison, with results still focused on sustainability as the lead heading, highlights the weakness in the actual data analysis portion. The themes generated do not correspond to the respondent comments "nobody to give them stipend." If lack of funds or limited mobilization or outreach were the reasons for example, the advocacy or demand generation programs were not sustained, then highlight this, rather than having the results in the same manner as the description of the NURHI program. More data analysis is required given the authors use of the qualitative research and I highly recommend using a standard text or article on what qualitative data analysis entails with revisions. Overall, the results section were poorly written with respondents quotes that do not correspond to the themes generated or rather the NURHI program described again in the results.

Response: We have reviewed the results as suggested and significantly revised the results section. We have also read through the illustrative quotes again to ensure that they correspond to the themes.

Discussion: This section started to describe sustainability. The authors should have started their paper with this definition as it would have helped to address some of their weakness. In fact, they should go back and use the papers on their reference list, from #28-33 and 35-38 as a guide for their revisions.

Response: Thank you for this useful suggestion. We have moved this section to the introduction and revised the discussion section

Reviewer: 2

I have heard about this study to detect whether NURHI had any sustained effects beyond the funding period(s), and believe that such findings are important for our field. This particular qualitative submission, however, does not sufficiently answer the question. What's needed is a more clearly designed way to establish a contrast between the responses from Ilorin (where NURHI activities were not continued in Phase II of the project), and the responses from Kaduna (where NURHI activities were continued in Phase II of the project). This contrast is the basis for detecting differences in responses, yet many of the qualitative observations - though informative - were not presented within a comparative "between-group" context, as you might expect in an evaluation framework.

Response: We have revised the results to better reflect the differences. Our analysis was limited by the responses provided and in many instances, some of the respondents in Kaduna had similar reports as their counterparts in Ilorin.

Moreover, the reader expects to see observations from a third city, Jos, where NURHI was never implemented. But while Jos is mentioned at the outset, it never appeared in the results. Such observations from a "control" site like Jos would add to the descriptive and explanatory findings.

Response: The findings focused on sustainability of the NURHI intervention activities which were facility-based or carried out by the facility staff. Hence, facility-level qualitative data were not collected from Jos and therefore this additional comparison is not possible.

More attention to structuring the responses within a comparative framework would be useful. Otherwise, though unintentional, the reader is likely to interpret each of the qualitative findings as an independent observation, rather than an observation that is dependent on group membership (i.e. belonging to Ilorin, to Kaduna, or to Jos).

Response: Thank you for this comment. We have significantly re-structured this aspect of the results and believe that it better reflects the comparison of findings between the two sites

At times it was not clear whether the respondents were qualified to speak of NURHI effects beyond the service provision & supply dimension of the project, because the understanding is that most respondents were service providers. Therefore it isn't clear whether their remarks about advocacy or demand effects were as valid as, for example, focus group discussions from the community might have been. A small note on testimonials: though the rigor and integrity paid to translation are commendable, often the testimonials were difficult to read and interpret.

Response: We appreciate that this is a limitation. The respondents however alluded to the effects of the mobilization in improving demand. They also noted the decrease in demand following close out of NURHI. We have included a statement about this as a limitation.

The note on testimonials has been taken and we have reviewed the responses to make them reader-friendly without losing the "voices" of the respondents.

NURHI also significantly changed its focus from its initial funding phase to its follow-up funding phase. The project shifted from a focus on the "urban poor" among six cities, to a focus on both urban and rural populations (poor?) in 3 States. For sustainability evaluation, this shift is problematic, because one would assume that during the 2nd phase of the project the units of analysis are harder to maintain, and establishing evaluative equivalence between the two phases becomes more difficult. How this affects the study design and the research questions is not clear.

Response: Thank you for this insightful comment.

We recognize that there was some modification in the focus of the intervention between the NURHI-1 and NURHI-2 and this could have made it difficult to evaluate sustainability of NURHI 1. We believe that this could be a problem for the NURHI-2 evaluation (and not the NURHI-1 evaluation) if it is not taken into consideration in developing the units of analysis for the NURHI-2 evaluation.

The change in focus could affect the NURHI 1 evaluation in terms of participants not being able to delineate which activities were NURHI-1 and NURHI-2. In order to minimize this, we clarified this difference to participants in the NURHI-2 site and emphasize that the questions were based on NURHI-1 interventions. We have included this as a limitation.

Speaking of research question, the reader is faced with at least two ways to observe "sustainability." The first is whether implementation of NURHI led to a sustainable impact on FP outcomes. The second is whether any aspects of NURHI implementation - any of its activities - continued after the project was closed. The two are not the same, but both types of sustainability effects are found in the observations and in the results. Can these two different indicators of sustainability be differentiated more clearly? Perhaps, but they are certainly both important, and the paper reveals promising findings in both of these areas.

Response: We appreciate these comments and suggestions and have described the aspects of sustainability that formed the focus of the paper. We have also clarified the change in focus of the program coverage between NURHI-1 and NURHI-2.

With these things in mind, what's missing is a clear hypothesis for how a major project like NURHI would "continue" (both in terms of impact and activities) after donor funding is terminated. Factors that contribute to continuation are discussed, but the discussion is not framed as a hypothesis that can be supported (or not) by the resulting qualitative findings.

For example, one hypothesis might be that when NURHI demonstrated treatment effects and efficiencies in activities that were priorities for the GON FMOH or MOH programs, they would be more likely to persist (in some form) beyond the life of the donor project. That indeed might be the case when the paper discussed the apparent sustainability of FP's integration with MNCH and HIV activities, because the GON has prioritized an integrated PHC approach nationwide, etc.

Response:

Thank you for the comment. Since these are qualitative data, we explore questions related to how things changed and perspectives of providers on these changes since NURHI-1 ended. We provide research questions rather than hypotheses and feel this is more transparent for a qualitative paper.

For the most part, "sustainability" of NURHI was not confirmed in many respondent testimonials because stipends were no longer available to have workers do their jobs. But this is obvious. What the reader wants to know is how to design a series of project-based interventions that contain specific mechanisms for maximizing the possibility that project activities and effects will be "taken-up" after the project ends.

What are these mechanisms? Training, for example, is often discussed as a priori possessing more of a "sustainability" dimension, because a project focused on training can be viewed as contributing to human capital development, and therefore to system strengthening.

In this context, the section on the NURHI 72-hour clinic makeover activity was very interesting. Was it a health system-strengthening activity? Was it sustainable? If no, why not? This activity is often cited as leading to substantial gains in modern contraceptive prevalence, but from this paper the reader is hoping to learn - more conclusively - whether the 72-hour clinic makeover was (1) designed with sustainability in mind; and (2) achieved sustainability. The reader is close to getting an answer, but not a complete one.

Response: These issues have now been discussed under the subsection, "sustainability of quality FP service provision" of the results as well as in the discussion.

Finally, the authors present a good list of recommendations for sustainability. However, the list is not a logical and direct extension of the observations contained in the body of the paper i.e. the list does not appear to be directly informed by and resulting from the way in which the qualitative findings were presented.

Response: The recommendations have been incorporated into the discussion section and have been revised to reflect the results.

Reviewer: 3

Overall, this research is interesting and fairly important, but there are some items that should be addressed before it is considered for publication.

It would be useful to have more information about the sampling approach (pg. 6). What is meant by "informed by data obtained from the quantitative phase"? Were all facilities from the first stage chosen, or only a subset? How were participants selected, and how was the number of participants determined?

Response: Additional details have been added on the selection of the subset of facilities included for the qualitative data collection.

Since there were other qualitative approaches used by the study, why does this manuscript report only results from the in-depth interviews, leaving out the focus groups and KII? The results seem incomplete without the other qualitative data.

Response: Thank you for this comment. We have improved the presentation of the findings. The current paper focuses on sustainability of the NURHI facility-based interventions. We felt that these would best be reported by the facility staff as they are in a position to provide information on how things were before, during and after the NURHI intervention.

It would be useful to have a more detailed description of the qualitative methods in the abstract.

Response: We have reviewed the abstract; however, we are constrained by the word limit stipulated by the Journal.

The manuscript would benefit from a thorough review of grammar, formatting, and spelling. For example, there are statements/phrases such as "over-generalization of the findings", "Demand for family planning services generated by NURI activities were...",

Response Thank you for highlighting these errors. We have reviewed the paper to ensure that typographical and grammatical errors are corrected.